# GRAPH4: A Security Monitoring Architecture Based on Data Plane Anomaly Detection Metrics Calculated Over Attack Graphs

Giacomo Gori , Lorenzo Rinieri , Amir Al Sadi , Andrea Melis , Franco Callegati and Marco Prandini *

Department of Computer Science and Engineering (DISI), Alma Mater Studiorum— Università di Bologna, 40136 Bologna, Italy; g.gori@unibo.it (G.G.); lorenzo.rinieri@unibo.it (L.R.); amir.alsadi@unibo.it (A.A.S.); a.melis@unibo.it (A.M.); franco.callegati@unibo.it (F.C.)
* Correspondence: marco.prandini@unibo.it

**Abstract:** The correct and efficient measurement of security properties is key to the deployment of effective cyberspace protection strategies. In this work, we propose GRAPH4, which is a system that combines different security metrics to design an attack detection approach that leverages the advantages of modern network architectures. GRAPH4 makes use of attack graphs that are generated by the control plane to extract a view of the network components requiring monitoring, which is based on the specific attack that must be detected and on the knowledge of the complete network layout. It enables an efficient distribution of security metrics tasks between the control plane and the data plane. The attack graph is translated into network rules that are subsequently installed in programmable nodes in order to enable alerting and detecting network anomalies at a line rate. By leveraging data plane programmability and security metric scores, GRAPH4 enables timely responses to unforeseen conditions while optimizing resource allocation and enhancing proactive defense. This paper details the architecture of GRAPH4, and it provides an evaluation of the performance gains it can achieve.

**Keywords:** P4; attack graphs; anomaly detection; security metrics; entropy

## 1. Introduction

Today's technological world of ubiquitous connectivity as well as the related intrinsic dependence on digital infrastructure has drastically increased the attention toward cybersecurity. Cyberattacks have evolved in sophistication and scale, thereby posing significant threats to individuals, organizations, and entire counties. Furthermore, ransomware attacks, data breaches, and Distributed Denial of Service (DDoS) attacks have become commonplace with far-reaching consequences. Amidst this escalating threat landscape, anomaly detection has become a crucial tool in cybersecurity arsenals. At its core, anomaly detection empowers organizations to identify deviations from normal system behavior which often imply intrusions or malicious activities. Hence, unlike signature-based approaches that rely on known attack patterns, anomaly detection strategies should perform well when unexpected threats happen [1].

Anomalies, however, are not always attacks. When this is the case, it is nontrivial to decide what is happening. As sustainability is becoming an essential objective to reach in any kind of system, especially energy-intensive ones, automation in the detection of and reaction to attacks is key to designing secure and optimized systems.

Thus, the accurate security evaluation of monitored data is pivotal to pinpoint which portions of our system are vulnerable and need to be fixed. We argue that measuring security can help us assess the actual level of system trustworthiness, which is otherwise hidden inside the architecture.

In our approach, security metrics are the core of measurable security. In fact, they are designed to help assess the overall security posture, identify vulnerabilities, track security incidents, and make informed security decisions [2]. Additionally, we foster a culture of continuous improvement, thus facilitating data-driven decision making to optimize security measures and fortify the organization's defense against evolving cyber threats.

Designing metrics is nontrivial and requires an in-depth knowledge of the domain we are analyzing, as we argued for Cyber–Physical Systems (CPSs) [3]. Moreover, metrics can be based on simple operations on low-level data, can be derived from models, or can be more complex when involving inferences.

Software-Defined Networks (SDN) is a paradigm that perfectly suits the purpose of security metrics. SDNs centralize the management of network resources in the network control plane, thereby allowing for the dynamic and automated control of network traffic, routing, and configuration. This allows for deep traffic inspection and simple data collection in real time and thus eases the definition and collection of metrics [4]. Unfortunately, metric calculation could demand high power consumption [5], and—especially in complex networks and critical infrastructure—this would aggravate the computational load on the network controllers implementing the control plane, thus leading to inadequate and unacceptable overhead [6].

Countermeasures are achievable by offloading monitoring tasks to the nodes in the data plane, which is possible if Data Plane Programmability (DPP) is available. DPP is the enabler of the solution we propose, i.e., the data plane can be programmed to gather information about the traffic or to process it. An example of offloading computation to the data plane to detect DDoS attacks is proposed by Ding et al. [7].

A metrics-oriented architecture to detect network anomalies using DPP is proposed by Vissicchio et al. [8]. In this work, the switches autonomously push alerts to the controller if an anomaly is detected. Then, the collected data are elaborated on the control plane. However, these solutions entail a sensible computational load on switches, thereby resulting in added forwarding delay that can lead to network saturation.

The solution presented in this work, GRAPH4, aims at overcoming this problem by requiring in-network computation only for the nodes that really may be at risk of a given attack. Attack graphs (AG) are generated by the control plane according to the active topology, and they are used to identify the vulnerable nodes. From the AGs, it is possible to know in advance what portion of the network needs to be monitored. This information is offloaded in the form of network rules to the programmable data plane components, which will then alert the control plane if an anomaly is spotted.

GRAPH4 can ensure a robust and resilient security posture, thereby allowing for the prompt mitigation of security incidents, thus optimizing and distributing the computation of the metrics. Leveraging switch programmability based on security scores allows for timely reactions to unexpected conditions: this dynamic approach optimizes resource allocation, minimizes response times, and fosters proactive defense. We demonstrate how GRAPH4 can save a big part of the network computation using simple AGs, especially for computationally expensive metrics.

The main contributions of this paper are as follows:

- We present a security monitoring architecture that integrates attack graphs generated on the control plane with anomaly detection metrics calculated on the data plane.
- We demonstrate how the control plane can exploit its knowledge of the network configuration in order to optimize the allocation of low-level metrics collectors, thus placing them only on specific data plane devices.
- We evaluate the effectiveness of GRAPH4 presenting a PoC related to Distributed Denial of Service detection.

The paper is organized as follows: in Section 2, we expose the concepts that we used in our work, in particular security metrics and DPP with a focus on the P4 programming language. In Section 3, we proceed to review the related literature. In Section 4, we propose the GRAPH4 architecture explaining its main components and features. In Section 5, we

provide a proof of concept on a real use-case scenario with DDoS attacks as an example, thereby showing the improvements that we can obtain with respect to existing solutions. In Section 6, we discuss the limitations of the study. Lastly, Section 7 includes our conclusion and proposals for future works.

## 2. Background

In this section, we describe the concepts and tools that are used to implement the GRAPH4 system. We start with security metrics, followed by a brief description of the programmable switches exploiting the P4 programming language, and conclude by recalling the definition of network traffic entropy, which is used to detect anomalies.

### 2.1. Security Metrics

Security metrics are indicators capable of presenting a level of security about aspects of the systems in terms of qualitative or quantitative assessment. The metrics that are suitable for specific environments can vary a lot [3]; therefore, in order to adopt the appropriate ones, it is crucial to make a selection of them based on a clear and unified view of the complete set. To assist in the process of justified decisions, we need a taxonomy of security metrics.

The literature classifies security metrics according to given categories. NIST suggests a taxonomy based on three main groups [9]: (i) management, which provides a high-level view of a security program's effectiveness and alignment with business objectives, (ii) operational, which focuses on the processes, procedures, and activities related to managing and maintaining security, and (iii) technical, which is usually measured and analyzed by IT and security teams. In this work, we will focus only on technical metrics, as we focus on network attacks.

Within the technical domains, metrics can be grouped according to different criteria. A very common approach in the literature is the one cited also in [5] that classifies technical metrics into four types:

- Vulnerability metrics, which consist of measures of system vulnerabilities that depend on the user, the interface, the hardware, or software in general. Examples are password [10], attack surface [11] and software vulnerabilities (https://nvd.nist.gov/vuln-metrics/cvss, accessed on 1 October 2023).
- Defense metrics, which measure the strength and the effort needed for placing defense mechanisms in a system. They consist of the evaluation of preventive, reactive, and proactive defenses, such as in [12].
- Situation metrics, which focus on the security state of a system and depend on the moment of analysis; these metrics dynamically evolve from time to time following the outcome of attack–defense interactions. Examples are metrics based on the frequency of security incidents or related to the investment for the security improvement [13]. Those are mainly subdivided into data-driven metrics, such as the network maliciousness metric [14], or model-driven metrics, such as the fraction of compromised computers.
- Attack metrics, which measure the strength of sustained attacks. The categories previously proposed focus on the assessment of the level of security of the system via the analysis of configurations and devices; conversely, attack metrics focus on quantifying and analyzing cyberattacks and threats faced and are essential for assessing risk, measuring the success of security measures, and guiding the allocation of resources. Examples are the network bandwidth that a botnet can use to launch DOS attacks or the occurrence of obfuscation in malware samples [15].

More ways to classify metrics exist, which are based on their scope (e.g., network, device, system, architecture, etc.) or the result types (e.g., quantitative or qualitative). Metrics evaluation can also be automated or manually configured: they can change dynamically at runtime or whenever a system reconfiguration occurs. Moreover, metrics can be simple measurements or be modeled as a complex system.

Without an assessment of its correctness, any kind of metric can lead to information that does not provide accurate and meaningful insights. Assessing the correctness and soundness of security metrics is crucial because it enables informed decision making, effective risk management, and sensible resource allocation. Accurate metrics help in evaluating the effectiveness of security measures, benchmarking against industry standards, and identifying improvement areas. They enhance transparency, build trust, and ensure the organization adapts to emerging threats appropriately. Thus, avoiding incorrect metrics is essential to prevent misinterpretations and misguided security actions. To ensure that a security metric is correct and sound, it is recommended to verify the reliability and relevance of the data sources used to calculate the metric, to evaluate the integrity and quality of the data to avoid potential biases or inconsistencies, to ensure that the metric is measurable and quantifiable allowing for easy comparison and analysis, to check the metric consistency and stability over time and under similar conditions, to analyze the metric within the context of the organization's overall security program for better understanding, to avoid metrics that incentivize undesirable behavior or manipulation, and to continuously review and update metrics to reflect changes in the threat landscape and security strategies [16].

Modern security metrics must be able to provide a holistic view of security, considering the interaction between different components of the system and how an action or event at one point may affect other parts of the infrastructure. Effectively addressing the complexity of threats requires the use of integrated metrics that can cover multiple aspects and provide a more comprehensive perspective of vulnerabilities and potential threats. The composability of security metrics allows for more agile adaptation to new and emerging threats and enables the optimization of their composition and sum [3].

### 2.2. P4 and Data Plane Programmability

P4 [17] is an open-source programming language that controls data plane packet processing. As depicted in Figure 1, a P4-enabled switch differs from traditional ones in two aspects: (i) the switch functions are not hardwired but specified by a P4 program, and (ii) control and data plane communication take place through a fixed-function device channel, but the data plane APIs are established by the P4 program. The Southbound Interface APIs that expose the particular features and protocols supported by the data plane are built using the specifications provided by P4Runtime [18] for abstracting the hardware interfaces. P4's primary goals are as follows:

- Reconfigurability: The packet parsing logic and processing rules can be dynamically installed and updated by the controller.
- Protocol independence: By giving rule names, key types, and typed match+action tables in the header fields, the controller can determine how to process header fields, freeing the switch from fixed actions taken on standard packet formats.
- Target independence: The P4 code is completely portable for every target. The P4 compiler is tasked with translating program features that make use of target-specific capabilities.

P4 is designed around an abstract model that explains the traffic forwarding of the switch through match+action steps that are organized in series, parallel, or both. The parser, which extracts header information and serves as a programmable interpreter of supported protocols, first handles inbound packets. The match+action tables, which choose the egress port and queue for the packet, obtain the extracted header data next. The packet may be sent, replicated, dropped, or cause flow control depending on the ingress processing. A P4 program defines the following elements to express the behavior of the data plane:

1. Header types: packet header definitions, i.e., the set of fields and their sizes.
2. Parsers: finite-state machines that map packets into headers and metadata.
3. Tables: data structures defining matching fields and actions applied to them.
4. Actions: code fragments that describe packet manipulation and can consider external data supplied by the control plane at runtime.

5. Match–action units: elements that construct lookup keys from packet fields' metadata and use them to find the right action and execute it.
6. Control flows: imperative blocks that describe packet processing on a target using the data-dependent sequence of match–action unit invocations.

Thus, P4 enhances traditional SDNs (heavily reliant on the OpenFlow protocol [19]) by addressing two well-known shortcomings [20]: (i) a significant communication overhead is generated between the data plane and the control plane and (ii) significant processing capabilities are needed at the controller.

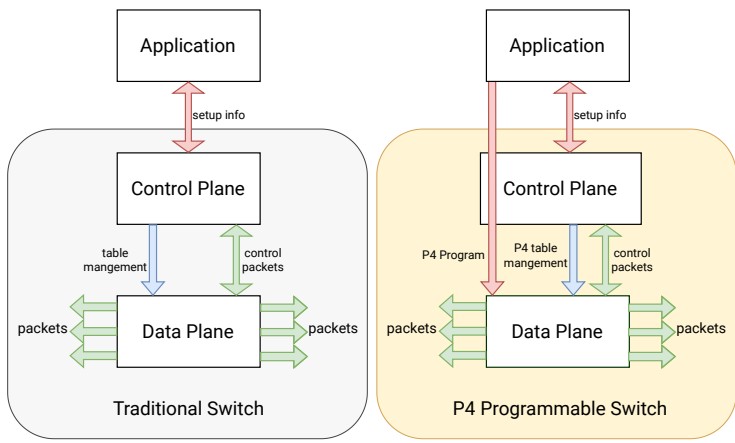

**Figure 1.** Comparison of the architecture of a traditional switch and a P4 programmable switch. The forwarding behavior of the P4 programmable switch can be configured directly from the application, setting the P4 program, while in a legacy switch, the application will determine the behavior of the control plane that will then configure the data plane.

P4's unrivaled expressiveness provided a revolutionary new perspective on network programmability and monitoring: with P4-programmable switches, it is instead possible for network operators to partially overcome such drawbacks. Programmable switches can execute a portion of network monitoring and security operations directly in their data plane pipeline and provide partially or fully processed information to the control plane. On the other hand, data plane programming has some intrinsic entry barriers, such as the need for enabled hardware and the effort needed for network architects to become proficient in the art of creating efficient and portable code [21].

The expressiveness limitations of OpenFlow-based SDNs do not allow coding algorithms on the data plane, such as network entropy calculation, which is presented in the next section. In fact, OpenFlow only allows for a fixed set of configurations, while P4 can be used to code custom packet processing algorithms.

*2.3. Network Traffic Entropy*

Entropy was introduced as a measure of uncertainty by Shannon in 1948 [22]. Assuming that $X$ is a dataset with a finite number $n$ of independent symbols, represented by $x_1, x_2, \ldots, x_n$, with corresponding probabilities $p = p_1, p_2, \ldots, p_n$, the entropy of $X$ is defined as follows:

$$H(X) = -\sum_{i=1}^{n} p_i \log p_i \qquad (1)$$

The entropy value ranges between 0 and $\log n$, reaching the upper bound when $X$ has a uniform distribution. To make entropy values independent of the number of distinct symbols, entropy can be normalized to vary from 0 to 1 as follows:

$$H_N(X) = \frac{H(X)}{\log n} \qquad (2)$$

From the above definitions, it is possible to define network traffic entropy as an indication of traffic distribution across the network [23]. Each network switch can evaluate the traffic entropy related to the network flows that cross it in a given time interval $T_{int}$ as:

$$H = -\sum_{i=1}^{n} \frac{f_i}{|S|_{tot}} \log_d \frac{f_i}{|S|_{tot}} \tag{3}$$

where $f_i$ is the packet count of the incoming flow $i$, $|S|_{tot}$ is the total number of processed packets by the switch during $T_{int}$, $n$ is the overall number of distinct flows and $d$ is the base of the logarithm. Network traffic entropy reaches its minimum value $H = 0$ when in the given time interval $T_{int}$, all packets $|S|_{tot}$ belong to the same flow $i$, while it reaches its maximum value $H = \log_d n$ when each of the $n$ flows transports only one packet.

## 3. Related Works

In the literature, security metrics that are proposed and related to network analysis focus especially on attack graphs: they are graphical representations of potential attack paths and the various steps an attacker might take to compromise a target system or network. They provide a visual depiction of the relationships between different vulnerabilities, system components, and attack techniques that an adversary could use to achieve their objectives. Over time, various research works exploited their potential. Wang et al. [24] proposed a general framework for designing network security metrics based on AGs. In [25], Lippmann et al. propose the Network Compromise Percentage Metric while evaluating the so-called defense-in-depth strategy using AGs. In [26], Mehta et al. compute a ranking of states in an AG based on the probability of attackers reaching each state during a random simulation; the PageRank algorithm is adapted for such a ranking; a key assumption made in this work is that attackers would progress along different paths in an AG in a random fashion. In this work [27] from Pamula et al., attack trees are replaced by attack trees with more advanced AGs and attack paths with attack scenarios. In [28], Leversage et al. proposed a mean time-to-compromise metric based on the predator state-space model. Homer et al. [29] address several important issues in calculating such metrics including the dependencies between different attack sequences in an AG and cyclic structures in such graphs. Poolsappasit et al. [30], instead, proposed Bayesian networks to quantify the chances of attacks and to develop a security mitigation and management plan as a metric.

More recent works start to consider vulnerabilities inside the nodes of the graph to make more accurate calculations: Wang et al. [31], instead of attempting to rank unknown vulnerabilities, propose a metric that counts how many such vulnerabilities would be required for compromising network assets: a larger count implies more security because the likelihood of having more unknown vulnerabilities available, applicable, and exploitable all at the same time will be significantly lower; Zhang et al. [32] presented a biodiversity-inspired metric based on the effective number of distinct resources with the idea that the larger the diversity in components, the more secure the system is because it is less probable that the same vulnerability is shared between different manufacturers; Ramos et al. [33] proposed metrics that consider the length and the number of attack paths resulting from the graph. An implementation of AGs very common in the literature is MulVAL [34], which is a project that since 2006 proposed an end-to-end framework and reasoning system that produces AGs by conducting multihost, multistage vulnerability analysis on a network. A recent work by Stan et al. [35] that uses this tool on modern network architectures leverages MulVAL to model multiple attack techniques such as spoofing, man-in-the-middle, DDoS, and other types of attacks.

Metrics that pertain to this category require a high-level view of the network in order to produce and use an AG. Moreover, many of them are resource intensive and cannot be computed on devices that must maintain a very low overhead, such as switches. A device such as a switch could implement only metrics of measure type but can still exploit the possibility of managing the data plane. The aforementioned studies all employ AGs as a

means of obtaining a comprehensive overview of potential attack vectors within network environments with the intent of deriving relevant metrics from such data. Distinguishing our research from these prior works, we extend our analytical scope beyond the metrics attainable through AGs alone. Instead, we integrate these metrics with lower-level data extracted from different vantage points. Furthermore, our investigation stands apart from previous efforts in its specific emphasis on networks featuring DPP and the exploitation of its intrinsic functionalities.

In contrast, considering low-level metrics deployable on network devices, the first contribution is the direct detection of DDoS attacks within the data plane of switches. However, the widely adopted data plane programming language, P4, lacks support for many arithmetic operations, limiting the straightforward implementation of advanced network monitoring functionalities required for DDoS detection. To address this limitation, Ding et al. [7] present two novel strategies for flow cardinality and normalized network traffic entropy estimation, which rely solely on P4-supported operations and ensure low relative error. Building upon these contributions, the authors propose a DDoS detection strategy based on variations of normalized network traffic entropy. The results demonstrate comparable or higher detection accuracy compared to state-of-the-art solutions while being simpler and executed entirely in the data plane. Moreover, Gao et al. [8] propose an alternative solution, as mentioned in the Introduction, allowing it to overcome the use of sketches and enabling switches to alert the controller automatically upon the detection of anomalies. They implement statistical checks in P4 by revisiting the definition and computation of statistical measures and collecting the techniques in a P4 library. Considering these works, the distinction inherent in our research lies in an opposite concept to what was previously discussed. Here, we consolidate the calculations of low-level metrics with those of high-level metrics, capitalizing on this integrated analysis within distinct layers of security evaluation.

## 4. GRAPH4 Architecture

As highlighted in the Section 2, the scientific literature proposed valuable ideas pertaining to security metrics calculation over a P4-based data plane to detect networking attacks—e.g., traffic entropy. However, the metric computation can introduce forwarding overhead that, in environments requiring strict latency or traffic shape requirements, is non-negligible.

GRAPH4 encompasses the combination of low-level metrics on the data plane with high-level ones on the control plane to optimize resource allocation while minimizing network computation. We envision a network architecture that, leveraging AGs, is able to pinpoint the vulnerable regions of the topology and protect them from possible attacks.

The control plane generates an AG, which detects all the possible paths that lead to a vulnerable host. With this information, the control plane is able to install the entropy calculation code only on the nodes that pertain to the vulnerable paths. Hence, we drastically reduce the number of network nodes that calculate the metric and thus the overall network overhead.

The architecture of GRAPH4 is shown in Figure 2, spanning across the two levels of the network: the control plane and the data plane.

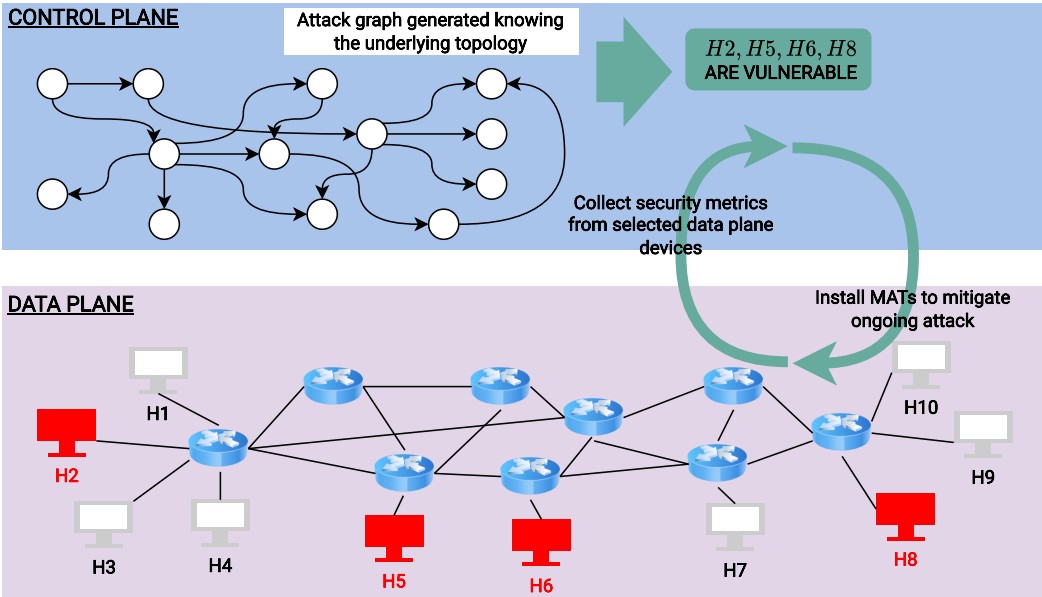

**Figure 2.** How the interaction between the control plane and the data plane works in GRAPH4. The hosts in red are the ones deemed vulnerable by the AG: the control plane instruments the switches to gather metrics on the switches close to those hosts.

### 4.1. Control Plane

In the control plane, the controller is in charge of generating an AG for the nodes in its network. To achieve this, it needs to provide two pieces of information to the AG generator: the description of the network topology and its hosts, with any active online services, and the target, which is the attack vector leading to the anomaly. With these elements, the generator provides the graph as output. With the AG, the controller achieves a twofold gain:

1.   It can calculate metrics over the AG to show high-level information on the security state of the network. Example metrics that are evaluated on AGs are the Standard Deviation and Mode of Path Lengths, the Number of Attack Paths [33] or the Defense Depth [36].
2.   It knows the set of nodes that are vulnerable to certain types of attack and can use this information to optimize the allocation of low-level metrics.

In this work, we will focus on the latter aspect.

### 4.2. Data Plane

In the data plane, on the other hand, the architecture follows a similar method to the one proposed by Vissicchio et al. [8]: the analysis is performed on the entire network. In our proposal, the analysis is carried out only over traffic concerning the vulnerable hosts identified by the controller.

When abnormal entropy levels are detected, the switches directly alert the controller, which is then able to take a decision and further actions, such as blocking the malicious traffic.

### 4.3. Workflow

To have a more in-depth view of how GRAPH4 works, we can summarize the workflow in six main steps that are shown in Figure 3. The first four steps are executed at the startup of a network:

1.   The AG Generator is fed with information about the topology of the network (hosts, active services, connections, etc.), which is used to generate the AG that identifies the vulnerable paths of the network.

2.　　The control plane, with that knowledge, can identify the vulnerable hosts that need to be monitored.

3.　　The control plane identifies the switches that are involved in forwarding traffic to or from said hosts and prepares the rules to be installed.

4.　　The Match Action Tables (MATs), which implement the low-level detection of attacks, are installed in the relevant switches.

Then, two more steps are repeated in a loop at network runtime:

5.　　The data plane carries out network monitoring by inspecting the traffic and, if the metrics indicate that an attack is detected, it notifies the control plane.

6.　　When the control plane is notified of an attack, it can deploy countermeasures, e.g., by installing additional MATs on the switches.

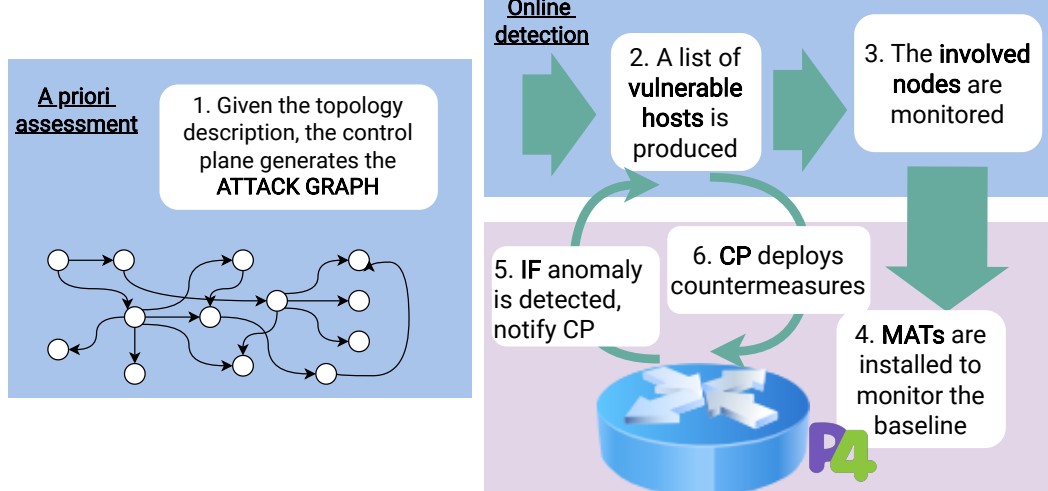

**Figure 3.** The six steps show the workflow of the GRAPH4 architecture between the control plane, which is represented as the zone with a blue background, and the data plane, with a violet background.

This process could be generalized to leverage the use of any metrics capable of providing indicators on attacks of different kinds. In fact, GRAPH4 is a control-data plane collaboration scheme revolving around two core features: (i) vulnerability assessment via AG and (ii) minimal but reliable distributed in-line attack detection. This idea shows an example of how the integration of model-based and measure-based metrics allows for the optimization and distribution of resource allocation—and ensuing energy usage—across the network, which is pivotal for in-line detection of attacks.

## 5. Proof of Concept

In this section, we present a Proof of Concept (PoC) to demonstrate the feasibility and effectiveness of GRAPH4 in detecting a Distributed Denial of Service attack. This attack targets one or more network hosts by firing a distributed volumetric attack that crosses multiple regions of the network with the goal of exhausting their band capacity and ultimately isolating them from the network. To detect these attacks, we need to deploy detection probes on the network; our goal is to make them as efficient as possible and to place them only where strictly needed to minimize computational and communications overhead.

The PoC illustrates how GRAPH4 can efficiently enhance network security by providing a holistic view of vulnerabilities and potential attack paths while monitoring and reacting in case of a detected attack. In terms of specific implementation choices, it combines high-level metrics on the controller by generating a MulVal AG to pinpoint the

vulnerable hosts and P4NEntropy [7] on the data plane as a low-level metric to calculate the normalized entropy and detect ongoing anomalies.

### 5.1. The Data Plane Low-Level Metric

The adopted low-level metric, P4NEntropy [7], estimates the normalized network traffic entropy in a given time interval, as described by Equation (3) in Section 2. It is implemented in P4, thus overcoming the lack of this language which does not natively support basic yet relevant arithmetic operations such as division, logarithm, and exponential function calculation as well as any operation on floating numbers or for loops.

P4DDoS is an application developed by the authors of [7] to trigger an alarm when the entropy metric P4NEntropy reaches a value crossing a given threshold. We modified the P4DDoS application to be compatible with GRAPH4 so as to be able to install the detection rules only for the vulnerable hosts that result from the AG.

The DDoS attack is detected when the P4NEntropy metric value drops below the adaptive threshold: as network traffic varies over time, the adaptability of the threshold is essential to avoid false positives. Figure 4 shows the detection of a DDoS attack in our simulation: when the metric value decreases under the threshold, initially set to 0.5, the controller installs MATs to block the malicious flow. As soon as the new rules are activated, the malicious traffic is automatically dropped by the switches, as proven by the return of the metric to its normal value. In our simulations, these actions take an average of 2.5 s from the detection of the attack to the installation of the new rules.

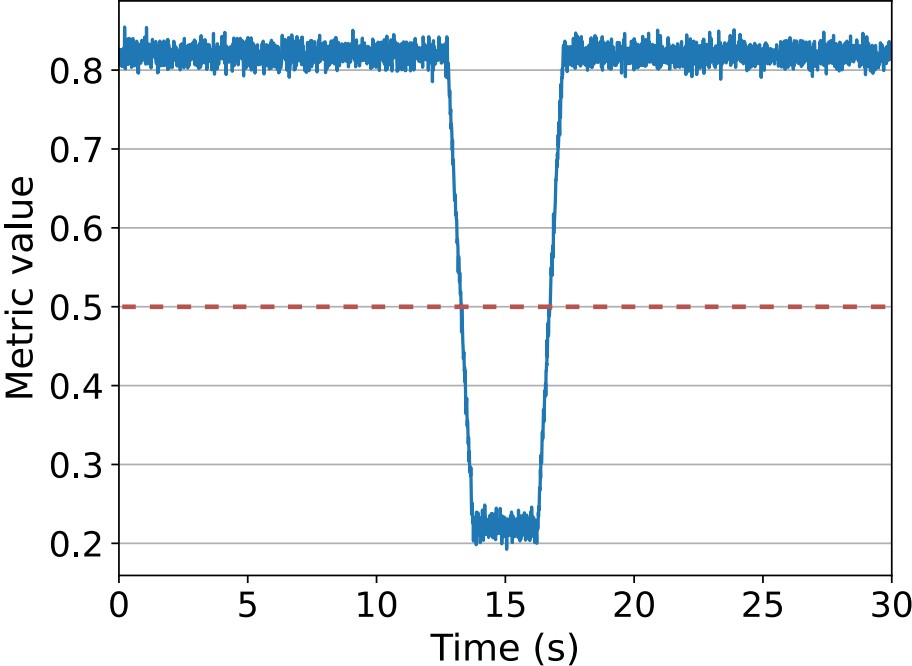

**Figure 4.** The variation of the P4NEntropy metric value during time in our tests: as it drops below the threshold of 0.5, the DDoS attack is detected and new rules are installed in the switch. Hence, the malicious traffic is blocked and the metric returns to its typical value.

### 5.2. AG Cuts

We emulate the network topology shown in Figure 5 using Mininet (https://mininet.org/, accessed on 1 October 2023) with a single P4 switch. The data plane is described by P4 code compiled using the bmv2 behavioral model (https://github.com/p4lang/behavioral-model, accessed on 1 October 2023), while the control plane is implemented as a custom controller we developed separately [21]. We abstract external hosts as a single traffic source connected to the P4 switch, which is also connected to three different subnets, each one

composed of six hosts with the role of destinations. Virtual links bandwidth is bounded by CPU capacity. We tested the topology on a Ubuntu 20.04 LTS Server with 14 GB of RAM and 3 CPU cores KVM machine.

In our simulation, to produce the AG, we use the extended version [35] of the popular MulVAL application to make it compatible with our use case. The knowledge that MulVAL uses to produce the graph is based on the assertion of some predicates that are written in the Datalog language [37].

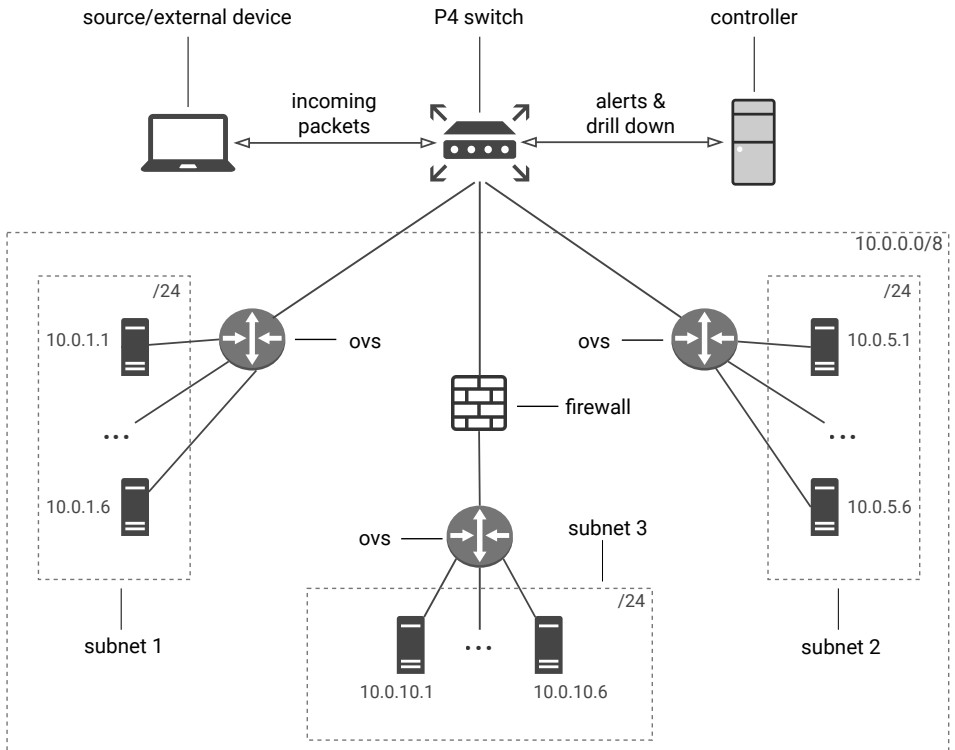

**Figure 5.** Our use-case test topology, based on the one used in [8], with another subnet (10.0.10.0/24) protected by firewall.

We produced an `input.P` file that describes the topology of our network presented in Figure 5. To express the various fundamental network components such as hosts, gateways, active services, and known vulnerabilities, we employed the extended primitive rule set presented in [35]. The Datalog content of the `input.P` file is shown in Figure 6: for the sake of brevity, in the figure, we replaced long lists of similar lines with comments explaining what has been omitted. With this input, MulVAL automatically generates the AG, trying to reach the goal written in the `input.P` file by verifying the predicates of his knowledge. Figure 7 shows the attack graph that is produced.

```
1    /* attacker info */
2    attackerLocated(internet).
3    malicious(attacker).
4
5    /* network topology */
6    isGateway(switchP4,subnetP4).
7    isGateway(ovs1,subnet1).
8    isGateway(ovs2,subnet2).
9    isGateway(ovs3,subnet3).
10
11   located(ovs1, subnetP4, ipSubnet).
12   located(ovs2, subnetP4, ipSubnet).
13   located(ovs3, subnetP4, ipSubnet).
14
15   located(host1, subnet1, ipSubnet).
16   /* repeat for every host of subnet 1, 2 and 3 */
17
18   hacl(internet, host1, tcp, 80).
19   /* repeat for every host */
20
21   /* active services */
22   networkService(host1, ssh, tcp, 80, _).
23   /* repeat for every host */
24
25   aclH(host1, _, _, host1, tcp, 80).
26   /* repeat for every host of subnet 1 and 2 */
27
28
29   /* vulnerability information */
30   vulHostDos(host1).
31   /* repeat for every VULNERABLE host */
32
33   attackGoal(dos(attacker,host1)).
```

**Figure 6.** The code of `input.P`, which is the input for MulVAL to generate the AG.

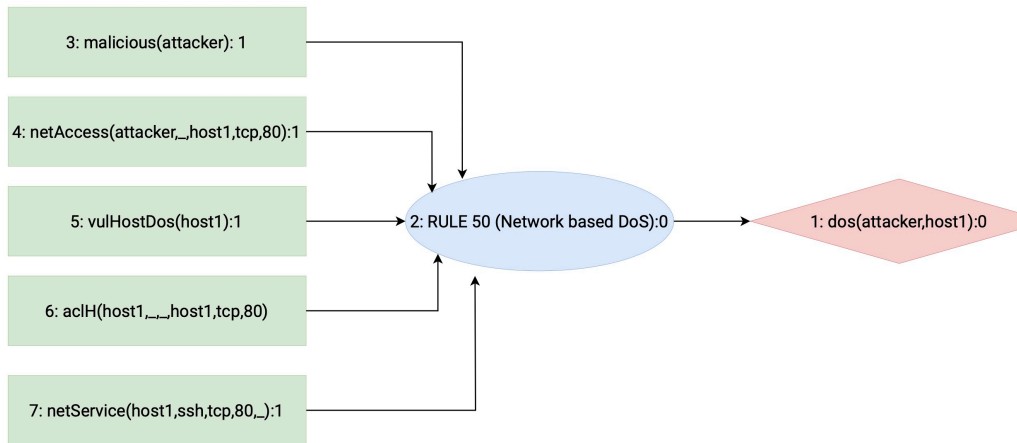

**Figure 7.** The AG generated from the topology shown in Figure 5 that refers to just one host of the vulnerable subnets. The graph is repeated for every host that pertains to the vulnerable subnet. Based on the Datalog code that defines the relationships between elements of the network and attacks, the graph shows all the conditions that need to be verified to complete the hypothetical attack goal—in this case, a DoS attack.

As a result, the vulnerable hosts are identified: they are the hosts that appear along a path in the AG; Figure 7 shows for the sake of clarity the AG for one host; the complete output would contain the AGs involving all hosts specified in the input file. Consequently, the $N_h$ hosts in the network are split into two sets, the $V$ vulnerable hosts and the $S$ non-vulnerable ones:

$$N_h = V + S \tag{4}$$

The switch will compute security metrics just for the traffic flows that have a vulnerable host as an endpoint. The metrics computation is performed over the switch and will increase the overall Packet Processing Time (PPT). The absolute computation time will obviously depend on the capacity of the available hardware so, to keep the discussion as general as possible, here we assume it is a multiple of a constant time unit $U_c$.

The security metric is computed using Equation (3). Packet Processing Times (PPTs) (The PPT is calculated as follows: we used Wireshark (https://www.wireshark.org/, accessed on 1 October 2023) to capture a packet timestamp $t_{in}$ at the ingress interface of the switch and the timestamp $t_{out}$ at the egress interface when the same packet is forwarded to the destination host. The packet processing time is then calculated by $t_{out} - t_{in}$.) are plotted in Figure 8. The blue color is used to plot the PPT of normal packets, i.e., not requiring metric computation because they cannot be part of the attack; the orange color is used to plot the PPT of packets used to calculate the security metric. The test is always performed by forwarding a series of 10,000 packets through the switch between the source and destination. The red lines correspond to the respective averages of the two sets of PPTs.

We plotted five sample cases, ranging from no packets undergoing security metric calculation up to security metric calculation for all packets, with intermediate cases of 12.5%, 25%, and 50% of packets triggering the security metric calculation. We believe this figure provides some important information and insight into the problem we are considering:

1. The PPT is indeed quite variable but shows statistical stability when comparing single values with the average; therefore, we assume we are facing a stationary and ergodic stochastic process;
2. The average PPT for the two cases is almost independent of the number of packets involved in security metric calculation, and because of this, we assume that we can make some general statement not depending on the actual shape of the AGs but just on the number of hosts identified as vulnerable;
3. The overall processing burden in the switch is a linear function of the number of traffic flows considered suspicious (i.e., the number of suspicious hosts), and therefore, we are interested in the security metric calculation. On average, the overall processing burden in the switch is a multiple of the number of flows according to the average packet processing time.

Based on the aforementioned considerations, we are now interested in understanding how much the data plane of the network is loaded by the additional task of security metric calculation in the switches. Let us call the total computation time consumed in the switch, $T_c$ in the following, as an indication of such overall load. Based on the discussion above, we assume $T_c$ is the product of the number of flows to monitor, which is the same as the number of vulnerable hosts $N_h$, for the average processing time of the involved packets that we estimated in Figure 8. This estimation on a single switch is in line with the demonstration of [8] Vissicchio et al., leaving the multi-switches case for future works.

As a result, we can extrapolate the following:

1. The overhead generated in a single switch by the process of metric elaboration will be linearly influenced by the number $V$ of hosts identified as possibly vulnerable;
2. The total overhead generated in the network by the entire metric calculation processes will have a directly proportional relationship with the number of switches that compute the metric.

As a consequence, to minimize the number of traffic flows, the network has to focus on calculating the flow entropy that is meaningful to minimize the load on the network control plane.

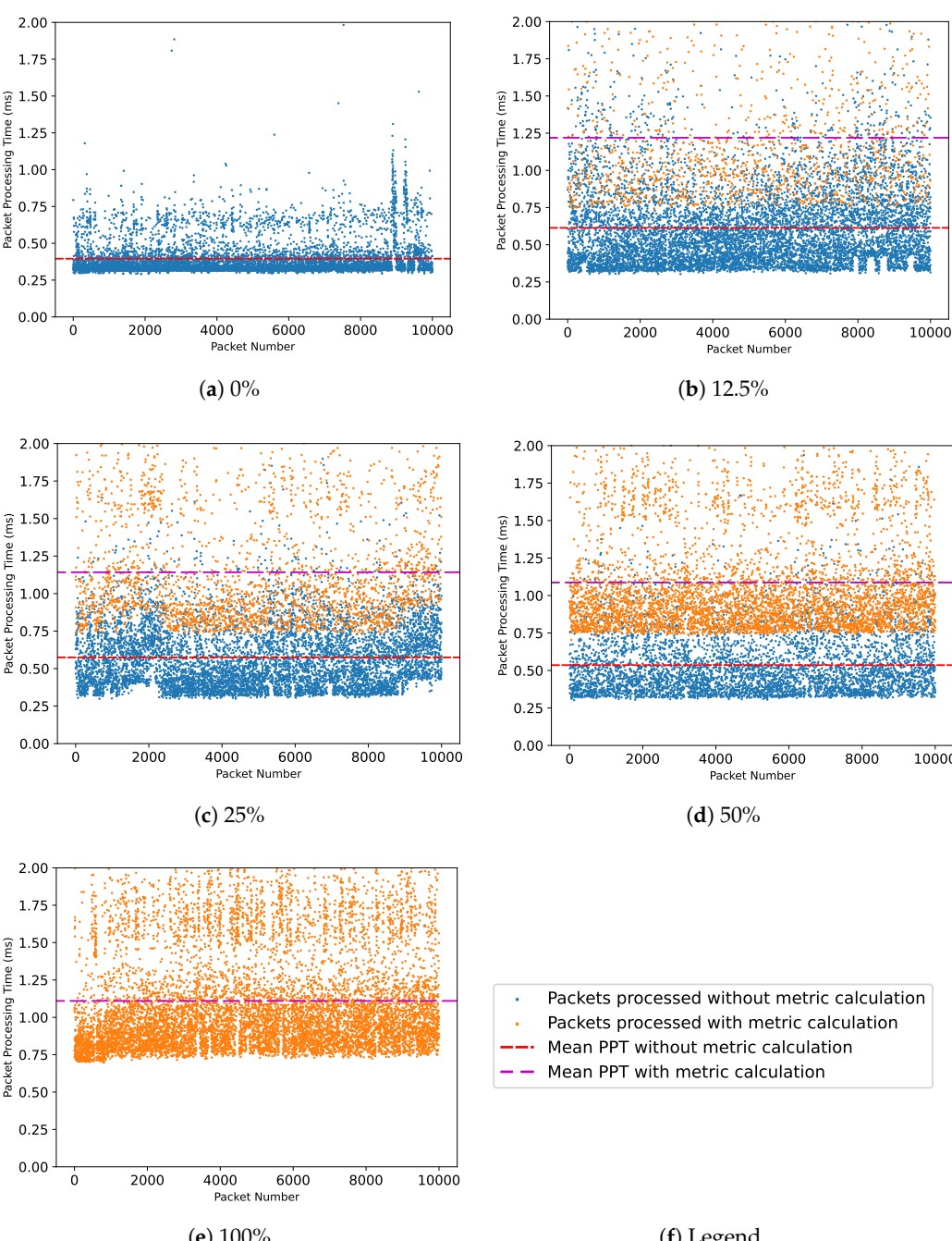

**Figure 8.** Packet Processing Time (PPT) in the switch plotted for 10,000 packets with varying percentages of metrics calculation. The goal of the figure is to show that the single PPT of a packet is independent and not influenced by the number of packets subject to metric calculation.

In practice, without GRAPH4, the $T_c$ value will be always proportional to $N_h$. Instead, with GRAPH4, this number is tied to $V$, which is by definition less than or equal to $N_h$. Therefore, if the network contains fewer vulnerable hosts, i.e., some hosts are not affected by DDoS attacks as proven by the AG, $T_c$ will be lower. In Figure 9, we summarize this concept: $T_c$ will be linearly increasing with the number of hosts in the network $N_h$ without

GRAPH4 ($V = N_h$). On the other hand, with GRAPH4, it will be upper bounded by the number of vulnerable hosts $V$, which is why the curves in Figure 9 become constant when $N_h = V$.

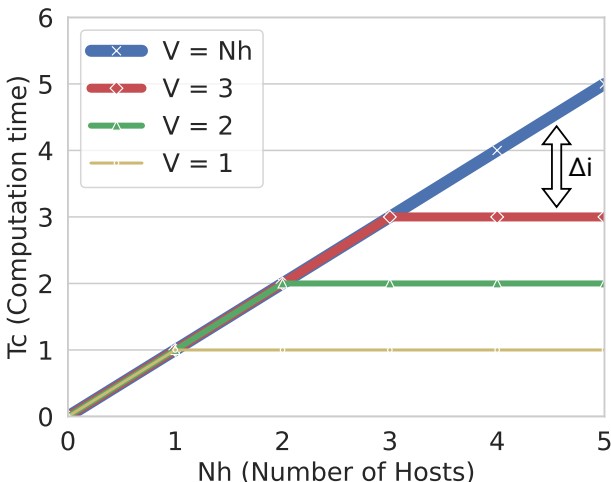

**Figure 9.** The Computation Time ($T_c$), measured in time units, based on the formulas shown before. The blue case is when $V = N_h$, and the function result is the same as without the use of AG. The other cases are with different values of $V < N_h$, i.e., there is only a minor subset of vulnerable hosts in the network: in those cases, we can see the improvement in the calculation time, which is the benefit that the use of GRAPH4 provides, and this is labeled as $\Delta_i$.

To give further evidence of the linear relationship between $T_c$ and $N_h$, we performed a series of five tests, each of them with 50,000 packets forwarded, within the same experimental environment as the test shown in Figure 8. We sent different amounts of packets directed to vulnerable hosts that need to be monitored from the switch: in the first case, 0% of the packets were directed to the vulnerable hosts, then 12.5%, 25%, 50% and 100%, mirroring the conditions of the previous experiment. During these tests, the mean value and variance of PPT were measured. The obtained results are depicted in Figure 10. The observed linear trend confirms our previous assumption on the independence between PPTs of packets within flows that are monitored or not by the switch for metric computation, thus confirming the linearity hypothesis.

Finally, we want to remark that the volume of additional data transferred between the control and data plane can be deemed negligible, since GRAPH4 only sends a small alert each time the entropy threshold is hit. In fact, an alert only contains the source and destination IP addresses of the flow hitting the entropy threshold, resulting in a minimal data payload of 64 bits per anomaly detection event.

Compared with traditional SDN-based mirroring approaches, where all the traffic is sent to the control plane, GRAPH4 only sends an alert every time an anomaly is detected. For example, high-speed networks often treat the Tbps of traffic: mirroring this volume of traffic to the control plane can result in low detection responsiveness and high detection overhead. On the contrary, GRAPH4 aggregates the attack in a 64-bit alert, which allows prompt control plane decisions.

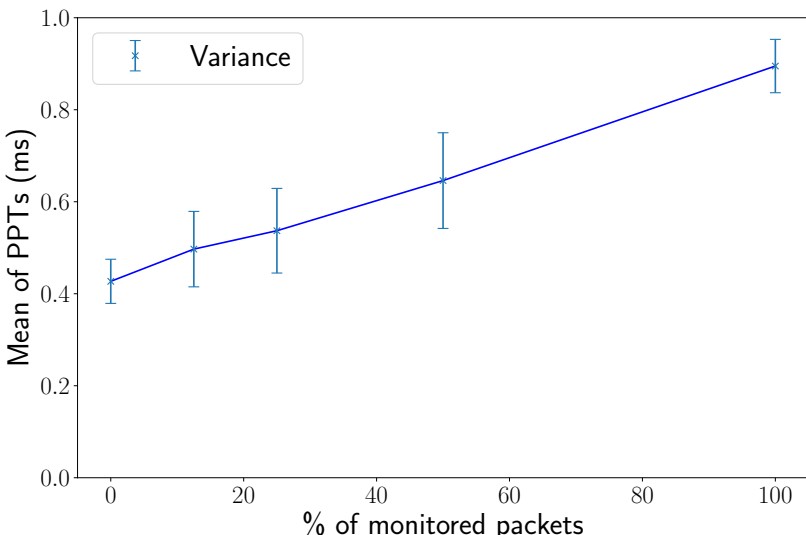

**Figure 10.** The mean and the variance of single-switch PPTs in five test scenarios, each one of them with 50,000 packets forwarded: from 0% of packets processed with metric computation to 100%.

## 6. Discussion and Limitations of the Study

The proposed approach shows promising features, yet it is important to acknowledge its open issues. One of the foremost limitations of the architecture lies in the completeness of vulnerability enumeration. In our approach, the AG generated on the controller relies on the information that describes the topology of the network as well as active services and their specific vulnerabilities. If any of that information is missed or not correctly identified, the AG may be incomplete. This incompleteness can lead to a significant loss of information, potentially resulting in a false sense of security. An attacker exploiting a vulnerability that was not accounted for in the AG would go undetected, posing a substantial security risk. For this reason, it is crucial to feed the AG generator with all the necessary information, thus imposing an a priori analysis of the network to recognize them.

Moreover, the workflow assumes a relatively immutable network environment. However, in some cases, network topology, hosts, and vulnerabilities can change dynamically and frequently. Handling these changes effectively and updating the graph in real time is a challenging task and is one of the main criticisms of AGs in general beyond our proposed use of them. Failure to adapt to dynamic changes could render the AG obsolete, leading to inaccuracies in threat detection.

Addressing these challenges and potential pitfalls is crucial to the successful implementation of the architecture in real-world network environments. Further research and development are needed to overcome these limitations and make the architecture more robust and effective in enhancing network security.

## 7. Conclusions and Future Works

In conclusion, our research has demonstrated the enhanced capabilities of real-time analysis with P4 through the integration of metric-based measurements. By employing a multi-level approach, we have successfully developed an architecture that comprises (i) AG generation on the control plane, to assess the vulnerable network end hosts, and (ii) at the data plane level, providing an optimized implementation of detection metrics and computation distribution.

Thus, this approach not only bolsters security but also contributes to the broader objective of sustainability by minimizing resource usage. In our PoC, we showcased the effectiveness of our methodology by utilizing MulVAL and entropy-based metrics such as P4NEntropy. The results of our experiments underscore the practicality and efficiency

of our approach in identifying and responding to security threats in real-time network environments while simultaneously promoting sustainable resource management.

As the field of network security continues to evolve, our research contributes to the ongoing efforts to guarantee network defenses and enhance the resilience of critical infrastructures in an environmentally responsible manner. We anticipate that our approach will find applications in a wide range of network security scenarios, offering a more efficient and targeted means of safeguarding digital assets against emerging threats while contributing to sustainability objectives.

Our study lays the foundation for future research directions in network security with security metrics. Future works could try to quantify, for some real network examples, the accurate enhancement of the total computation time for the metric thanks to the exemption of some switches from the monitoring. In addition, opportunities for innovation in this field will come from investigating alternative metrics within the architecture, other than the ones that we proposed, and delving deeper into diverse high-level and low-level metric integration. We are also aware of the need for further exploration of the dynamic adaptability of threat models, of real-world deployment scenarios, and of the interplay between security and sustainability goals. Lastly, integrating machine learning for advanced threat detection represents a promising avenue for enhancing network security and could be included in this architecture to increase accuracy in detection and adaptability to new threats. These future works aim to further advance the state of the art and contribute to more robust and resource-efficient security solutions.

**Author Contributions:** Conceptualization, G.G., L.R. and A.A.S.; Methodology, G.G., L.R. and A.A.S.; Testing G.G., L.R. and A.A.S.; Writing—original draft, G.G., L.R. and A.A.S.; Writing—review and editing, A.M., F.C. and M.P.; Supervision, A.M., F.C. and M.P. All authors have read and agreed to the published version of the manuscript.

**Funding:** This research received no external funding.

**Data Availability Statement:** Data are contained within the article.

**Acknowledgments:** This work was partially supported by project SERICS (PE00000014) under the MUR National Recovery and Resilience Plan funded by the European Union—NextGenerationEU; Giacomo Gori was supported by the National PhD in Cybersecurity (IMT Lucca); Amir Al Sadi was supported by the National Operating Program Research and Innovation (PON 2014–2020), action IV.5 PhD on Green subjects, by the Advanced Studies Institution (ISA) and by the International PhD College.

**Conflicts of Interest:** The authors declare no conflict of interest.

## Abbreviations

The following abbreviations are used in this manuscript:

| | |
|---|---|
| DDoS | Distributed Denial of Service |
| CPS | Cyber–Physical Systems |
| SDN | Software-Defined Networks |
| DPP | Data Plane Programmability |
| AG | Attack Graph |
| MAT | Match Action Table |
| PoC | Proof of Concept |
| PPT | Packet Processing Time |

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
