# Peer review of "GRAPH4: A Security Monitoring Architecture Based on Data Plane Anomaly Detection Metrics Calculated over Attack Graphs"

_futureinternet, doi:10.3390/fi15110368_

Round 1

Reviewer 1 Report

Comments and Suggestions for Authors

The authors present GRAPH4, a security monitoring architecture based on data plane anomaly detection metrics. The overall work is interesting. The methodology is explained well, while the paper is structured well. However, the paper can be further improved before publication. Particular recommendations are provided below.

- The contributions of the paper are not clear. The should be stated and described briefly in the introductory part.

- In section 3, the authors should include the differences and innovations of this work with respect to similar papers.

- In the background section, the authors should describe the choice of P4 instead of SDN-related solutions.

- The authors should enhance the simulation/numerical results of their work.

- Finally, the paper should be re-checked entirely for typos and writing issues.

Comments on the Quality of English Language

The paper should be re-checked entirely for typos and writing issues.

Author Response

We would like to thank Reviewer 1 for the valuable suggestions, and we would like to respond to the comments as follows.

COMMENT 1: “The authors present GRAPH4, a security monitoring architecture based on data plane anomaly detection metrics. The overall work is interesting. The methodology is explained well, while the paper is structured well. However, the paper can be further improved before publication. Particular recommendations are provided below.

  • The contributions of the paper are not clear. The should be stated and described briefly in the introductory part.

RESPONSE: Thanks for the comment, we outlined the main contributions of the article by modifying Section 1 (Introduction), highlighting the features of the architecture that we proposed: GRAPH4 exploits the security information obtained from the Attack Graphs to optimize the allocation of metrics computation on the data plane.

COMMENT 2: “In section 3, the authors should include the differences and innovations of this work with respect to similar papers.“

RESPONSE: We have addressed the issue, including the differences between similar papers and ours, with lines 275-282 and 305-309.

COMMENT 3: “In the background section, the authors should describe the choice of P4 instead of SDN-related solutions.“

RESPONSE: We provided a brief clarification on why traditional SDNs are not suitable to implement GRAPH4 on the data plane on lines 223-227. This is due to the limitations of OpenFlow, the standard Southbound Interface protocol to configure the data plane.  

COMMENT 4: “The authors should enhance the simulation/numerical results of their work.“

RESPONSE: We have added numerical results obtained from additional tests, including an analysis of packet processing time and its change as the percentage of packets monitored varies, as well as improved the exposition of results and simulation by adding graphs and descriptions throughout Section 5, PoC. We have also added details about our simulation environment at lines 417- 423.

COMMENT 5: “Finally, the paper should be re-checked entirely for typos and writing issues.“

RESPONSE: We carefully revised the manuscript to fix typos and all syntax and grammar errors.

Reviewer 2 Report

Comments and Suggestions for Authors

The proposed architecture and its research need to be described in more detail and better.

Is the goal to distribute the metric calculation processes over the network? How is it done?

In the proposed architecture GRAPH4 (system?), entropy is proposed to evaluate the metric. However, only the general entropy calculation formula is provided, and how are the entropy of various metrics (4 metrics) calculated in the proposed solution?

The proof of concept is weak.

The model is greatly simplified; it states:

“1. The overhead generated in a single switch by the process of metric elaboration will be linearly influenced by the number of hosts.

2. The total overhead generated in the network by the entire metric calculation processes will have a direct proportionality relationship with the number of switches that compute the metric.”

Are linear dependencies really everywhere? Please explain why.

Only time consumption is evaluated, and how other characteristics, e.g. costs of data transmission between nodes and others.?

The research results presented in Fig.8 do not appear reliable and realistic. This is a very simplified model, with linear dependencies everywhere.

Author Response

We would like to thank Reviewer 2 for the valuable suggestions, and we would like to respond to the comments as follows.

COMMENT 1: “The proposed architecture and its research need to be described in more detail and better.“

RESPONSE: We fixed the problem by explaining more clearly the composition of the architecture, in particular, we have added lines 322-328 to explain that the Attack Graph is generated by the Control Plane, which is able to provide all the network information relevant for the generation. With the Attack Graph, the Control Plane is able to install the entropy metric calculations in the Data Plane, only in the switches and for the hosts that are involved in the attack paths.

COMMENT 2: “Is the goal to distribute the metric calculation processes over the network? How is it done?“

RESPONSE: The lines added for the previous answer should explain better how the calculation of metrics is distributed across the network. However, We also have modified lines 329-361, with the goal of trying to clarify even more how it works, explaining with more detail the steps that occur: the goal is to distribute the metric computations over the network at Data Plane and Control Plane, by deploying an Attack Graph in the Control Plane, from which we can obtain metrics as shown in the Related works, Section 3, but also from which we can optimize the allocation of metric computations over the Data Plane, by deploying the computations only for those hosts that result to be vulnerable from the Attack Graph.

COMMENT 3: “In the proposed architecture GRAPH4 (system?), entropy is proposed to evaluate the metric. However, only the general entropy calculation formula is provided, and how are the entropy of various metrics (4 metrics) calculated in the proposed solution?“

RESPONSE: We clarify this aspect by modifying section 5, PoC. We explain that the metric that is calculated is just one, and it is based on the entropy calculation of Formula num. 3; this indicator is essential as a DDoS detection strategy based on variations of normalized network traffic entropy, as it is indicated in section 3, Related Works.

COMMENT 4: “The proof of concept is weak.“

RESPONSE: We have entirely reshaped the structure of the Proof of Concept trying to demonstrate its validity more clearly. In particular, we conducted further tests and presented the results by describing them, adding plots and providing reasons on why they support our hypothesis. Their details are explained in more depth in the next answers. Thus, we demonstrate how GRAPH4 can devise a holistic strategy to minimize the in-line detection overhead while promptly detecting DDos attacks. In addition, we outlined in the Introduction (Section 1) how the proposed architecture is evaluated with respect to the presented PoC.

COMMENT 5: “The model is greatly simplified; it states:

  • The overhead generated in a single switch by the process of metric elaboration will be linearly influenced by the number of hosts.
  • The total overhead generated in the network by the entire metric calculation processes will have a direct proportionality relationship with the number of switches that compute the metric.

Are linear dependencies really everywhere? Please explain why.“

RESPONSE: We have added a better explanation on the reason in Section 5, I will briefly summarize the content here. We perform some simulation in the environment presented in lines 417-423, considering the packet processing time (PPT) of the switch for every packet. In the first simulations, we sent 10000 packets, varying the percentage of packets directed to vulnerable hosts (i.e. packets of which we calculate the security metric, as they are directed to hosts that are vulnerable for the Attack Graph), from 0% to 100%. 

We have added Fig.8 to show the results that demonstrate that there is no correlation between PPT of packets to which we perform the security analysis and the ones that go directly to the destinations. For this reason, by changing the percentage of packets to be monitored, the PPT changes accordingly following a linear trend, as shown in figure 10, that we obtained by graphing (Fig.10) the average PPT time, and variance, in a series of tests in which we sent 50000 packets, varying the percentage of packets directed to vulnerable hosts as before.

COMMENT 6: “Only time consumption is evaluated, and how other characteristics, e.g. costs of data transmission between nodes and others.?”

RESPONSE: We have added Fig. 8 and 9 also to consider other aspects, that are data transmission costs in terms of packet processing time caused by entropy computations, showing statistical information and numerical results that explain better the delay introduced by the metric computations and the optimizations that our architecture propose, and also a consideration regarding the amount of data that needs be transmitted in lines 533-541.

COMMENT 7: “The research results presented in Fig.8 do not appear reliable and realistic. This is a very simplified model, with linear dependencies everywhere.“

RESPONSE: As explained in the previous answer, we showed the results of other tests that confirm our hypothesis of linearity, which nevertheless needs some assumptions, namely that the number of packets are equivalent for each host (lines 493-495), therefore the linear relationship is directly transmitted from the number of packets (already demonstrated) to the number of hosts. This assumption is intentional and justified by the purpose of the PoC, namely to demonstrate the improvement introduced by GRAPH4, and should be considered as a first step toward further investigation of this topic and its further study, which, also for pedagogical purposes, allows a simplified, but nevertheless valid, understanding of the improvement introduced.

Reviewer 3 Report

Comments and Suggestions for Authors

The submitted manuscript proposes a security monitoring architecture based on data plane anomaly detection metrics calculated over attack graphs.

Some comments are given below:

An interesting aspect would be the behavior of the architecture in unstable conditions. I mean, how how much unstability can be tolerated? Please clarify.

The presentation should be much improved. English writing style should be revised. For example, it is not common to sentences with such phrases as “Here [26], Mehta et al.” (line 219) or “[29], Homer et al.” (line 226).

Numbers lower that ten are usually written in words (see line 398).

Formatting should also be revised. For example, the title usually does not end with a dot.

Regarding the figures, they also need attention.

A cursor should be removed from Fig. 6

Time units are missing in Fig. 8.

Comments on the Quality of English Language

needs attention

Author Response

We would like to thank Reviewer 3 for the valuable suggestions, and we would like to respond to the comments as follows.

COMMENT 1: “The submitted manuscript proposes a security monitoring architecture based on data plane anomaly detection metrics calculated over attack graphs. Some comments are given below: An interesting aspect would be the behavior of the architecture in unstable conditions. I mean, how how much unstability can be tolerated? Please clarify.“

RESPONSE: Thanks to your comment, we clarified more the use case structure, underlying in a much direct way that the Proof of Concept (Section 5) analyzes the topic of Distributed Denial of Service detection. This is clarified at the start of Section 5. Thus, we demonstrate how GRAPH4 can devise a holistic strategy to minimize the in-line detection overhead while promptly detecting the targeted hosts.

COMMENT 2: “The presentation should be much improved. English writing style should be revised. For example, it is not common to sentences with such phrases as “Here [26], Mehta et al.” (line 219) or “[29], Homer et al.” (line 226).“

RESPONSE: We carefully revised the manuscript to fix all syntax and grammar errors and improved the English writing style. In particular, we rewrote the reported sentences employing a more common and formal style.

COMMENT 3: “Numbers lower that ten are usually written in words (see line 398).“

RESPONSE: We carefully revised the manuscript to check that all numbers lower than ten are written in words. However, we decided to leave in numerical forms the entropy ranges in Section 2.3 and the decimal numbers in Section 5.1.

COMMENT 4: “Formatting should also be revised. For example, the title usually does not end with a dot.“

RESPONSE: We removed the dot from the title, removed any unnecessary space between paragraphs, and corrected typo errors in formulas in Section 2.3.

COMMENT 5: “Regarding the figures, they also need attention. A cursor should be removed from Fig. 6. Time units are missing in Fig. 8.“

RESPONSE: We fixed the problem in Figure 6. Instead, in Figure 8, we are not providing specific time measurements, but rather illustrating the results of parametric calculations. Therefore, the graph represents values in terms of units of time, rather than precise and quantifiable time values. The focus of the graph is to demonstrate the relationship and trends based on various parameters, rather than providing exact time measurements. We try to clarify this aspect by modifying the caption of the figure.

Round 2

Reviewer 1 Report

Comments and Suggestions for Authors

The authors addressed all the comments. Therefore, the paper can be accepted for publication

Reviewer 2 Report

Comments and Suggestions for Authors

The authors took into account most of the notes.

Reviewer 3 Report

Comments and Suggestions for Authors

The paper is much better now. I am satisfied with the revision and have no other comments.